# Reactivities of Hydroxycinnamic Acid Derivatives Involving Caffeic Acid toward Electrogenerated Superoxide in *N*,*N*-Dimethylformamide

**Tatsushi Nakayama** [1,*] 🆔 **and Bunji Uno** [2]

[1] Department of Pharmacy, Gifu Pharmaceutical University, Daigaku-Nishi, Gifu 501-1196, Japan
[2] Faculty of Pharmacy, Gifu University of Medical Science, 4-3-3 Nijigaoka, Kani, Gifu 509-0923, Japan; buno@u-gifu-ms.ac.jp
[*] Correspondence: tnakayama@gifu-pu.ac.jp; Tel.: +81-58-230-8100

**Abstract:** Reactivity of (2*E*)-3-(3,4-dihydroxyphenyl)prop-2-enoic acid (caffeic acid), classified as a hydroxycinnamic acid (HCA) derivative, toward electrogenerated superoxide radical anion ($O_2^{\bullet-}$) was investigated through cyclic voltammetry, in situ electrolytic electron spin resonance spectrometry, and in situ electrolytic ultraviolet–visible spectrometry in *N*,*N*-dimethylformamide (DMF), aided by density functional theory (DFT) calculations. The quasi-reversible redox of dioxygen/$O_2^{\bullet-}$ is modified in the presence of caffeic acid, suggesting that $O_2^{\bullet-}$ is scavenged by caffeic acid through proton-coupled electron transfer. The reactivities of caffeic acid toward $O_2^{\bullet-}$ are mediated by the ortho-diphenol (catechol) moiety rather than by the acryloyl group, as experimentally confirmed in comparative analyses with other HCAs. The electrochemical and DFT results in DMF suggested that a concerted two-proton-coupled electron transfer mechanism proceeds via the catechol moiety. This mechanism embodies the superior kinetics of $O_2^{\bullet-}$ scavenging by caffeic acid.

**Keywords:** hydroxycinnamic acid; caffeic acid; superoxide radical anion; cyclic voltammetry; electron spin resonance spectrum; proton-coupled electron transfer

## 1. Introduction

Caffeic acid ((2*E*)-3-(3,4-dihydroxyphenyl)prop-2-enoic acid, $CafH_2(COOH)$) is a natural phenolic compound classified as a hydroxycinnamic acid (HCA) derivative. All plants produce caffeic acid as an intermediate in the biosynthesis of lignin, a principal component of woody plant biomass [1]. $CafH_2(COOH)$ delivers a surprising number of health benefits; for example, it mitigates carcinogenesis, neurodegeneration, and other age-related diseases, and induces pharmacological activities such as immunomodulatory and anti-inflammatory effects [2–5]. The carcinogenicities and anticancer activities of $CafH_2(COOH)$ have been enthusiastically researched but with mixed results [2,6]. $CafH_2(COOH)$ can target several diseases by scavenging the reactive oxygen species (ROS) generated in biochemical processes and pathologies. Therefore, ROS scavenging by HCAs involving $CafH_2(COOH)$ as an antioxidant might play a therapeutic role. However, whether $CafH_2(COOH)$ exerts a substantial effect on any human disease or delivers health benefits to humans remains unclear because solid evidence is lacking.

To confirm the medicinal effect of HCAs, especially of $CafH_2(COOH)$ that possesses antioxidant activity, we must elucidate the chemical reaction mechanism, including the electron transfer (ET) reaction that scavenges ROS such as superoxide radical anions ($O_2^{\bullet-}$, the precursor of the hydroperoxyl radical, $HO_2^{\bullet}$) and the hydroxyl radical ($HO^{\bullet}$). Clarifying the mechanism of ROS scavenging by HCAs is necessary for understanding the related medicinal effect, as the ROS generated around lesions and inflammatory organs are direct causes of several pathologies [4,6]. Previous studies in the field of electrochemistry have reported how HCA is oxidized by electrodes in aqueous or non-aqueous (aprotic)

solvents [7–9], where the reactivity of HCAs toward the ROS was imitated by the heterogeneous electrochemical process. In these studies, antioxidant activity of HCA was assessed using cyclicvoltammetry, although the electrooxidation behavior varies depending on the presence or absence of protons. In the pioneering work by Hotta et al. [9], a well-developed two-electron reversible redox of $CafH_2(COOH)$ in acidic-aqueous media was observed, demonstrating that protons from the aqueous solvent relate to the oxidation. Conversely, cyclic voltammograms (CVs) of $CafH_2(COOH)$ in an aprotic acetonitrile solvent demonstrated an irreversible oxidation, as shown in the pioneering work by Masek et al. [8]. These electrochemical results imply that the chemical reaction mechanism involving proton transfer (PT) and ET between HCAs and the ROS cannot be observed in aqueous solvents, because solvent-derived protons interfere with the PT from hydroxyl groups of HCA. As is widely known, effective antioxidant activity usually requires the presence of ortho- or para-diphenolic hydroxyl groups (OH) for quinone–hydroquinone $\pi$-conjugation. $CafH_2(COOH)$ belongs to the phenylpropanoids, possessing both an ortho-diphenol (catechol, $CatH_2$) moiety and an acryloyl group linked via a C6–C3 skeleton. Antioxidant studies of $CafH_2(COOH)$ employing numerous assays and methodologies have been published, such as 1,1-diphenyl-2-picryl-hydrazyl free radical scavenging, 2-azino-bis(3-ethylbenzthiazoline-6-sulfonic acid) radical scavenging, total antioxidant activity by the ferric thiocyanate method, and $O_2^{\bullet-}$ scavenging with metal chelating activities [2,3]. Meanwhile, computational studies have provided mechanistic insights into the structure–antioxidant activity relationships of HCAs [10–13]. These studies evaluated the thermochemical properties of HCAs, such as their bond dissociation energies of OH and ionization potentials, thereby clarifying the energetics of the ROS-scavenging reaction. A pioneering work by Nsangou et al. [14] investigated the antioxidant activity of $CafH_2(COOH)$ toward $HO_2^{\bullet}$ and $HO^{\bullet}$. Being more reactive than $O_2^{\bullet-}$, these radicals are substantial cellular toxins in the living body. From the structure–property relationships of ROS scavenging by $CafH_2(COOH)$, the activity was mainly contributed by the $CatH_2$ moiety, which has a high degree of conjugation and can delocalize $\pi$-electrons with a resonance structure [11,13]. The characteristic resonance effect of $CafH_2(COOH)$ is considered to thermodynamically stabilize its radical product. However, isolated $HO_2^{\bullet}$ and $HO^{\bullet}$ are highly reactive and their reactions are difficult to observe experimentally; consequently, conclusive evidence of the ROS-scavenging mechanism of $CafH_2(COOH)$ is lacking in the literature.

There are several possible mechanisms of ROS ($O_2^{\bullet-}$, $HO^{\bullet}$, and $HO_2^{\bullet}$) scavenging by HCAs: single ET, hydrogen atom transfer (HAT) involving proton-coupled electron transfer (PCET) [15–19], sequential proton-loss electron transfer [20], and superoxide-facilitated oxidation (SFO) [21–23]. Along the SFO mechanism, the initial PT from the substrate to $O_2^{\bullet-}$ generates $HO_2^{\bullet}$, and a rapid dismutation into hydroperoxide ($H_2O_2$) and dioxygen ($O_2$) follows. Then, the $O_2$ formed in the dismutation process oxidizes the substrate anion [24]. On the other side, the other mechanism involves direct ET. In an aprotic solution, various phenolic antioxidants, polyphenols [18], 1,2- and 1,4-benzendiols (catechol [16] and hydroquinone [17]), and monophenols including aminophenols [19,25,26], scavenge electrogenerated $O_2^{\bullet-}$ through the PCET characterized by quinone–hydroquinone $\pi$-conjugation. Although $O_2^{\bullet-}$ is not so electrophilic but a Brønsted base, $HO_2^{\bullet}$ formed through the protonation of $O_2^{\bullet-}$ is a strong oxidant with a short lifetime. Therefore, the initial PT and subsequent oxidation (ET) are closely related to each other through the PCET between $CafH_2(COOH)$/anion and $O_2^{\bullet-}$/$HO_2^{\bullet}$ that embodies the actual mechanism of $O_2^{\bullet-}$/$HO_2^{\bullet}$ scavenging.

In this study, we investigate the homogeneous chemical reactivities between electrogenerated $O_2^{\bullet-}$ and HCAs involving $CafH_2(COOH)$ using cyclicvoltammetry, in situ electrolytic electron spin resonance (ESR) spectral measurements, and in situ ultraviolet–visible (UV–vis) spectral measurements in *N,N*-dimethylformamide (DMF) solution. Avoiding the interference of solvent-derived protons, the ROS scavenging reaction by HCAs is inferred from the electrochemical measurements by applying density functional theory (DFT). In addition, this study provides mechanistic insights into the structural features of

$CafH_2(COOH)$, namely, the $CatH_2$ moiety and the acryloyl group for the PCET mechanism characterized by $\pi$-conjugated redox reaction. Accordingly, we present valuable information regarding $O_2^{\bullet -}$ scavenging by HCAs including $CafH_2(COOH)$, which is assumed to provide substantial health benefits as a phytoalexin.

## 2. Materials and Methods

### 2.1. Chemicals

Highest-grade (2*E*)-3-(3,4-dihydroxyphenyl)prop-2-enoic acid (caffeic acid, $CafH_2$ (COOH), >98.0%), (2*E*)-3-(4-hydroxy-3-methoxyphenyl)prop-2-enoic acid (ferulic acid, >99.0%), (*E*)-3-(3-hydroxy-4-methoxyphenyl)prop-2-enoic acid (isoferulic acid, >97.0%), ethyl (*E*)-3-(3,4-dihydroxyphenyl)prop-2-enoate (ethyl caffeate, Et-$CafH_2$, >95.0%), (*E*)-3-(4-hydroxyphenyl)-2-propenoic acid (*p*-coumaric acid, >98.0%), and sodium methoxide ($CH_3ONa$, 95.0%) were purchased from Sigma-Aldrich Inc. (Tokyo, Japan) and were used as received. DMF (spectrograde, 99.7%) was used as the solvent for electrochemical and electrolytic ESR/UV–vis spectral measurements, and ferrocene (Fc) used as a potential reference compound, were purchased from Nacalai Tesque Inc. (Kyoto, Japan) and used as received. Dinitrogen ($N_2$) gas (99.0%) and $O_2$ gas (99.0%) were obtained from Medical Sakai Co., Ltd. (Gifu, Japan). Tetrapropylammonium perchlorate (TPAP, >98.0%) used as a supporting electrolyte was purchased from Tokyo Chemical Industry Co., Ltd. (Tokyo, Japan) and prepared as described previously [27].

### 2.2. Cyclic Voltammetry and In Situ Electrolytic ESR/UV–Vis Spectrum Measurements

Cyclic voltammetry was performed in a standard three-electrode system: working electrode, a 1.0-mm-diameter glassy carbon (GC); counter electrode, a coiled platinum (Pt, 99.99%); reference electrode, a silver/silver nitrate ($Ag/AgNO_3$) (containing an acetonitrile solution of 0.1 mol $dm^{-3}$ tetrabutylammonium perchlorate and 0.01 mol $dm^{-3}$ $AgNO_3$). All measurements were performed at 25 °C using an ECstat-301 electrochemical analyzer (EC-frontier Co., Ltd., Kyoto, Japan) and electrochemical software (Supplementary Materials, Table S1). Before the experiments, the working electrode was polished with alumina paste on a polishing wheel, rinsed with deionized water and acetone, and dried. Calibration of the reference electrode was referenced to the ferrocenim ion/ferrocene couple ($Fc^+/Fc$), and all reported potentials are referenced to the potential of the $Fc^+/Fc$ couple. A JES-FA200 X-band spectrometer (JEOL Ltd., Tokyo, Japan) and an OCEAN HDX spectrometer (OptoSirius Co., Ltd., Tokyo, Japan) were used for ESR and UV–vis spectral measurements, respectively. The controlled-potential electrolysis was performed at room temperature in two types of cells: an in situ electrolytic ESR cell with a 0.5-mm-diameter straight Pt wire sealed in a glass capillary as the working electrode (5.0-mm-length), and an optically transparent thin-layer electrochemical (OTTLE) cell (path length: 1.0 mm) with a Pt-mesh working electrode (10.0 mm × 20.0 mm) sandwiched between two pieces of quartz glass (Supplementary Materials, Figure S1). The working electrodes (planar glassy carbon, Pt wire, and Pt mesh) were chosen because they do not cause side reactions such as electrode-absorption with solutions or substrates. Samples were prepared in a glove box completely filled with $N_2$ gas to prevent contamination by moisture. The DMF solutions were saturated with $O_2$ by air-bubbling the gas for ca. 2–3 min, and the gas was passed over the solutions during the measurements to maintain a constant concentration of $O_2$ at $4.8 \times 10^{-3}$ mol $dm^{-3}$.

### 2.3. Theoretical Calculations

All calculations were performed at the DFT level with the Becke three-parameter Lee–Yang–Parr (B3LYP) hybrid functional implemented in the Gaussian 16 Program package [28]. This functional was chosen because it obtains good geometries of the reactants, transition states (TS), and products in PCET reactions between phenolic compounds and free radicals [29]. The highest occupied molecular orbital (HOMO) and lowest unoccupied molecular orbital (LUMO) were computed from geometry optimization with frontier orbital

theory. In the calculations for geometry optimization, vibrational frequency calculation, intrinsic reaction coordinate (IRC) calculation, and population analysis of each compound, we employed the standard split-valence triple $\zeta$ basis sets augmented by the polarization 3df,2p and diffusion orbitals 6-311+G(3df,2p). The polarized continuum model (PCM) under the default settings of Gaussian 16 was used to estimate the solvent contribution of DMF to the standard Gibbs free energies, which is widely employed in thermodynamic characteristic studies of solvation. The zero-point energies and thermal correction, together with the entropy, were used to convert the internal energies to standard Gibbs energies at 298.15 K. The natural bond orbital (NBO) technique was used for electron and spin calculations in the population analysis [30].

### 3. Results and Discussion

*3.1. Cyclic Voltammetry of $O_2/O_2^{\bullet-}$ in the Presence of HCAs*

Reactivities of HCAs shown in Scheme 1 toward electrogenerated $O_2^{\bullet-}$ were demonstrated in the CVs of $4.8 \times 10^{-3}$ mol dm$^{-3}$ of $O_2$ in the presence of HCAs in DMF (Figure 1). In aprotic solvents such as DMF, the relationship between ET and PT is well demonstrated in electrochemical and electrolytic-spectral measurements, avoiding interference of solvent-derived protons. In the CVs, one-electron reduction of $O_2$ (Equation (1)) is demonstrated in a quasi-reversible redox reaction with the initial cathodic scan generating $O_2^{\bullet-}$ and following reoxidation to $O_2$ in the returned anodic scan (1c/1a, bold lines in Figure 1). The generated $O_2^{\bullet-}$ is lowly reactive toward aprotic DMF. The reversible CVs investigated here became irreversible in the presence of phenolic HCAs (Figure 1a–e) at various concentrations (0 to $5.0 \times 10^{-3}$ mol dm$^{-3}$). As the CVs of bubbled $N_2$ showed no peak over the potential range, the loss of reversibility in the CVs of $O_2/O_2^{\bullet-}$ was attributed to an acid–base reaction; specifically, the $O_2^{\bullet-}$ acts as a Brønsted base forming $HO_2^{\bullet}$ along the initial PT (Equation (2)).

**Scheme 1.** Structures of HCAs considered in this study. (**a**) (2*E*)-3-(3,4-Dihydroxyphenyl)prop-2-enoic acid (caffeic acid), (**b**) (2*E*)-3-(4-hydroxy-3-methoxyphenyl)prop-2-enoic acid (ferulic acid), (**c**) (*E*)-3-(3-hydroxy-4-methoxyphenyl)prop-2-enoic acid (isoferulic acid), (**d**) ethyl (*E*)-3-(3,4-dihydroxyphenyl)prop-2-enoate (ethyl caffeate), and (**e**) (*E*)-3-(4-hydroxyphenyl)-2-propenoic acid (*p*-coumaric acid).

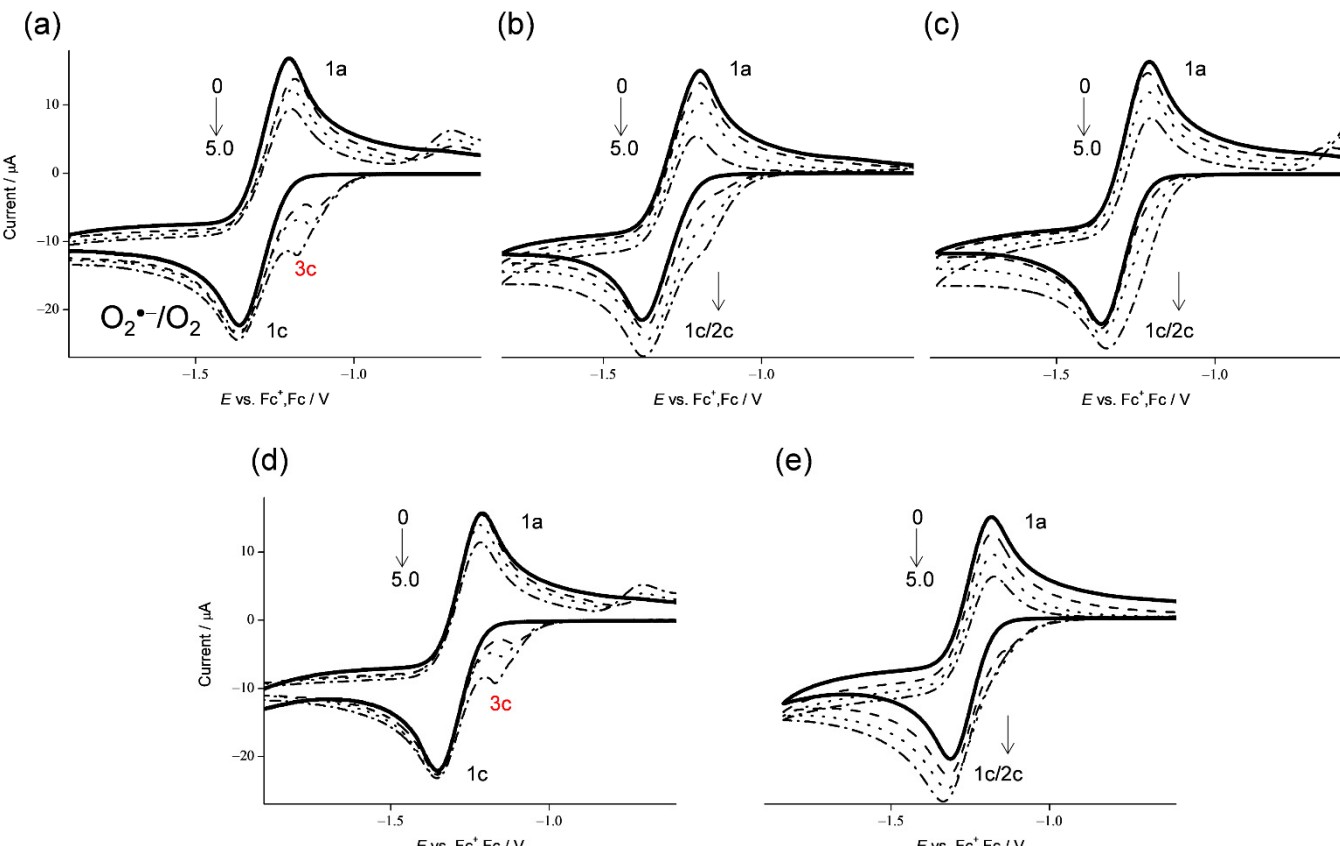

**Figure 1.** CVs of $4.8 \times 10^{-3}$ mol dm$^{-3}$ O$_2$ in the presence of (**a**) CafH$_2$(COOH), (**b**) ferulic acid, (**c**) isoferulic acid, (**d**) Et-CafH$_2$, and (**e**) *p*-coumaric acid, in DMF containing 0.1 mol dm$^{-3}$ TPAP. All CVs were recorded with a GC electrode at a scan rate of 0.1 V s$^{-1}$. Concentrations ($\times 10^{-3}$ mol dm$^{-3}$) are 0, 1.0, 3.0, and 5.0 (arrows indicate the concentration changes).

The initial PT and consequent reduction of HO$_2^\bullet$ (Equation (3)) gave rise to bielectronic CVs involving its cathodic current (2c) in the presence of ferulic acid, isoferulic acid, and *p*-coumaric acid (Figure 1b,c,e, respectively). This bielectronic CV did not appear in the presence of CafH$_2$(COOH) and Et-CafH$_2$ (Figure 1a,d, respectively) and was replaced by a prepeak (3c), demonstrating scavenging of HO$_2^\bullet$ by ET from the deprotonated anions (CafH(COOH)$^-$ and Et-CafH$^-$) (Equation (4)) forming substrate radicals (CafH(COOH)$^\bullet$ and Et-CafH$^\bullet$) and hydroperoxyl anion (HO$_2^-$). Consequently, a monoelectronic CV appeared.

$$O_2 + e- \leftrightarrow O_2^{\bullet -} \ (E^\circ = -1.284 \text{ V vs. Fc}^+/\text{Fc}) \tag{1}$$

$$O_2^{\bullet -} + \text{CafH}_2 \rightarrow \text{HO}_2^\bullet + \text{CafH(COOH)}^- \text{ (the initial PT)} \tag{2}$$

$$\text{HO}_2^\bullet + e^- \rightarrow \text{HO}_2^- \ (E^\circ = -0.4 \text{ to } -0.2 \text{ V vs. Fc}^+/\text{Fc}) \tag{3}$$

$$\text{HO}_2^\bullet + \text{CafH(COOH)}^- \rightarrow \text{HO}_2^- + \text{CafH(COOH)}^\bullet \text{ (ET)} \tag{4}$$

$$\text{HO}_2^- + \text{CafH(COOH)}^\bullet \rightarrow \text{H}_2\text{O}_2 + \text{Caf(COOH)}^{\bullet -} \text{ (the second PT)} \tag{5}$$

Based on the CVs, we rationalized that O$_2^{\bullet -}$ formation associated with PT from HCA leads to the irreversible bielectronic reduction of O$_2$ to H$_2$O$_2$ driven by the exergonic reduction of HO$_2^\bullet$/HO$_2^-$. Additionally, we divided the CVs of O$_2$/O$_2^{\bullet -}$ in the presence of HCAs (Figure 1a–e) into two typical curves: type A, an irreversible two-electron process observed in electro–chemical–electro reactions (Equations (1)–(3)); and type B, an irreversible one-electron process observed in electro–chemical–chemical reactions (Equations (1), (2), (4) and (5)) leading to the scavenging of O$_2^{\bullet -}$. Figure 2 summarizes Equations (1)–(5) showing

the electrochemical mechanisms of $O_2/O_2^{\bullet-}$ in the presence of (a) $CafH_2(COOH)$ and (b) ferulic acid.

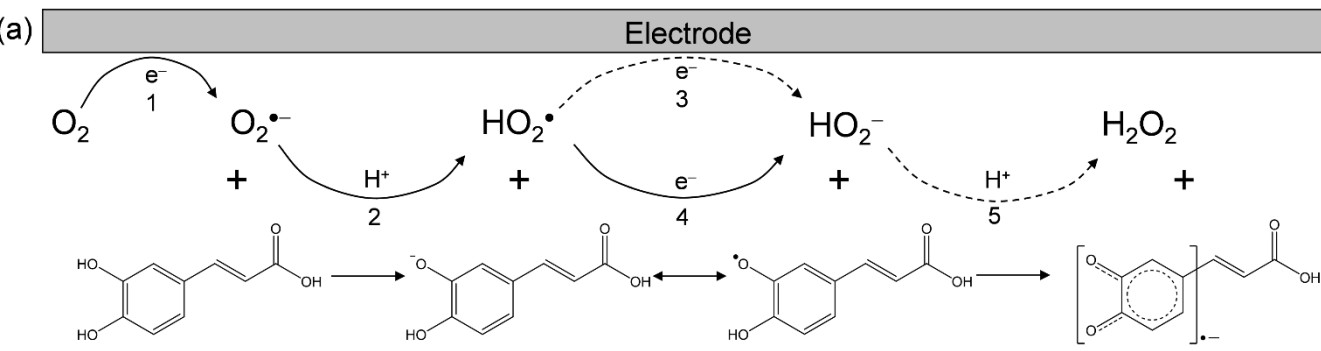

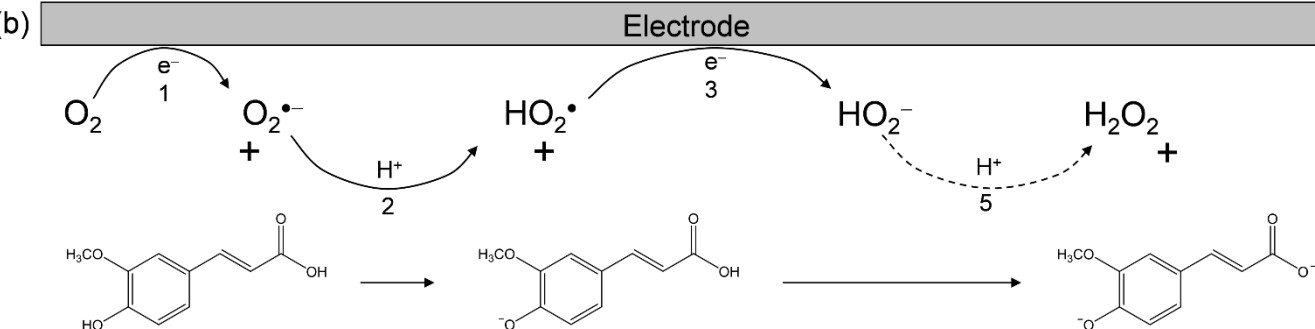

**Figure 2.** Electrochemical mechanisms of $O_2/O_2^{\bullet-}$ in the presence of (**a**) $CafH_2(COOH)$, and (**b**) ferulic acid in DMF. [1] one-electron reduction of $O_2/O_2^{\bullet-}$, [2] the initial PT from substrate to $O_2^{\bullet-}$, [3] exergonic reduction of $HO_2^{\bullet}/HO_2^{-}$, [4] ET from substrate anion to $HO_2^{\bullet}$, [5] the second PT to $HO_2^{-}$.

In this scenario, the CVs (Figure 1) recorded in the presence of ferulic acid, isoferulic acid, and *p*-coumaric acid were type A, demonstrating the absence of $O_2^{\bullet-}$ scavenging and the appearance of a cathodic-current of $HO_2^{\bullet}$. Conversely, the CVs in the presence of $CafH_2(COOH)$ and $Et$-$CafH_2$ were type B, demonstrating the scavenging of $O_2^{\bullet-}/HO_2^{\bullet}$. The prepeak in the CVs of type B is mainly ascribed to radical anions ($Caf(COOH)^{\bullet-}$ and $Et$-$Caf^{\bullet-}$), which are formed through the PCET from the $CatH_2$ moiety of the substrate. The slightly different cathodic prepeaks in Figure 1a,d can be explained by the reduction of $HO_2^{\bullet}$ via PT from the acryloyl group of $CafH_2(COOH)$. These CV results suggest a PCET reaction between $O_2^{\bullet-}$ and its $CatH_2$ moiety, which is independent of PT from the acryloyl group.

### 3.2. In Situ Electrolytic ESR/UV–Vis Spectral Analyses of $O_2/O_2^{\bullet-}$ in the Presence of HCAs

To confirm the $O_2^{\bullet-}$ scavenging by HCAs, the solutions of the CV experiment were analyzed by electrolytic ESR using an in situ cell (scanning time = 4 min) and by UV–vis spectral measurements using an OTTLE cell (Supplementary Materials, Figure S1). These spectra were acquired under a potential applied at $-1.3$ V, which corresponds to the electrogeneration of $O_2^{\bullet-}$ (Equation (1)). ESR spectra were obtained only in the presence of $CafH_2(COOH)$ and $Et$-$CafH_2$, suggesting that $O_2^{\bullet-}$ was scavenged in these cases (Figure 3a). An almost identical ESR spectrum was obtained in the presence of $Et$-$CafH_2$ (data not shown). In these ESR spectra, the fine coupling constants for hydrogen ($a_H$/mT) were assigned to hydrogens of $CatH_2$ ($H^a$, $H^b$, $H^c$: 0.23, 0.21, 0.19 mT) in the substrate radical anions ($Caf(COOH)^{\bullet-}$ and $Et$-$Caf^{\bullet-}$) formed through the PCET. This result demonstrates that radical spins are barely distributed on the carboxyl group of $CafH_2(COOH)$ and the ethyl ester group of $Et$-$CafH_2$. The in situ ESR system, which is

sensitive to hyperfine couplings, showed no clear splitting derived from acryloyl hydrogens ($H^d$, $H^e$: 0.01 mT).

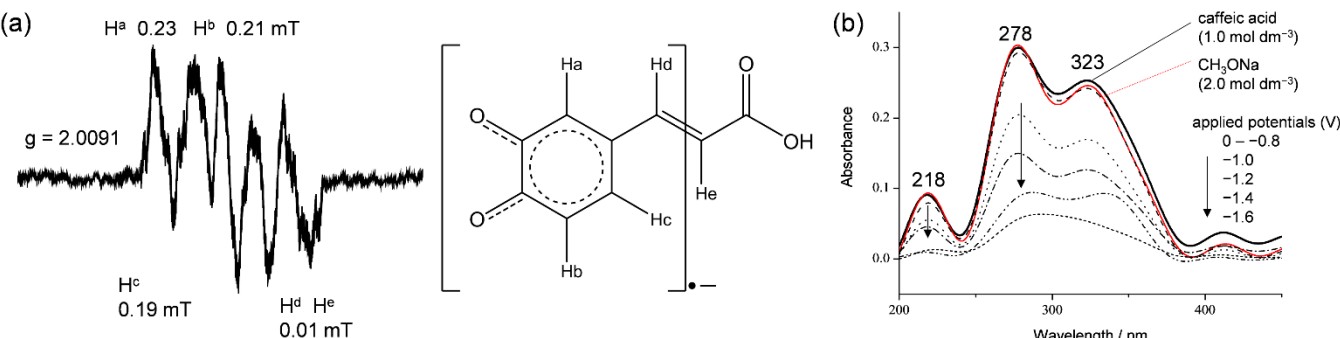

**Figure 3.** (**a**) ESR and (**b**) UV–vis spectra of a $O_2/O_2^{\bullet-}$ solution exposed to $CafH_2(COOH)$ in DMF under an applied potential of $-1.3$ V vs. $Fc^+/Fc$.

Figure 3b shows the in situ electrolytic UV–vis spectra of the CV solution containing $CafH_2(COOH)$ ($1.0 \times 10^{-3}$ mol dm$^{-3}$). These spectra were measured in the absence and presence of $O_2$ (in the former case, $O_2$ was removed by purging with $N_2$). The spectrum of $CafH_2(COOH)$ alone presented characteristic absorption bands centered at 218, 278, and 323 nm. This spectrum was unchanged under an applied potential of 1.0 to $-2.0$ V vs. $Fc^+/Fc$ without $O_2$ (data not shown), demonstrating that $CafH_2(COOH)$ was not electrolyzed without deprotonation. Additionally, the spectrum did not alter in the presence of $2.0 \times 10^{-3}$ mol dm$^{-3}$ $CH_3ONa$ without applying potentials (red line in Figure 3b). As $CafH_2(COOH)$ is deprotonated by the Brønsted base $CH_3ONa$, this spectrum was attributed to a mixture of $CafH_2(COOH)$, $CafH(COOH)^-$, and $CafH_2(COO)^-$. Under $O_2$ ($4.8 \times 10^{-3}$ mol dm$^{-3}$), the spectrum diminished at applied cathodic potentials above $-1.0$ V. From these spectral changes, the homogeneous reaction between $CafH_2(COOH)$ and the electrogenerated $O_2^{\bullet-}$ was inferred to involve the initial PT and the subsequent ET forming $Caf(COOH)^{\bullet-}$. This radical product was detectable in the in situ electrolytic ESR system, conversely undetectable in the UV–vis spectra, showing a decomposition of the radical product ($Caf(COOH)^{\bullet-}$). Analogously to the CV results, the spectra revealed that the initial PT (Equation (2)) and the following reactions involving ET between $HO_2^{\bullet}$ and $CafH(COOH)^-$ (Equation (4)) rapidly underwent base-catalyzed oxidation. The CV and spectral measurements imply that $O_2^{\bullet-}$ was successfully scavenged by $CafH_2(COOH)$ through the PCET over the time scale of the CV measurements.

### 3.3. DFT Optimization of the Stable Structure of $CafH_2(COOH)$ and Its Deprotonated Anion

To clarify the mechanism of the PCET between $CafH_2(COOH)$ and $O_2^{\bullet-}$ in DMF, we performed DFT calculations using the B3LYP hybrid functional employing the PCM method. First, the stable structures of $CafH_2(COOH)$ and the conformers of its deprotonated anions ($CafH_2(COO)^-$, $CafH(COOH)^-$) along the initial PT route were obtained from energy scanning of the dihedral angle around the acryloyl group. Figure 4 shows the optimized structures along with their calculated standard Gibbs free energy changes ($\Delta G°$/kJ mol$^{-1}$, 298.15 K) along the PT route. The distributed charges on the OH protons of $CafH_2(COOH)$ were obtained in an NBO analysis (Supplementary Materials, Table S2).

Comparing the charge distributions on the three OH protons of $CafH_2(COOH)$ (0.502 on 3OH, 0.507 on 4OH, and 0.502 on 1OH of the acryloyl group), one observes that the 4OH proton was more reactive in the acid–base reaction of $O_2^{\bullet-}$ than the other OH protons. The $\Delta G°$s of the initial PT also show the higher reaction feasibility of $CafH(COOH)^-$ ($-8.95$ kJ mol$^{-1}$) than of $CafH_2(COO)^-$ (5.81 kJ mol$^{-1}$). According to these calculation results, PT is initiated at the OH of the catechol moiety, although the 3OH and 4OH are indistinguishable along the dihedral rotation. The electrochemical results indicate the

occurrence of PT from the three OHs of CafH$_2$(COOH), but only 3OH and 4OH of the CatH$_2$ moiety are involved in the O$_2{}^{\bullet-}$ scavenging reaction.

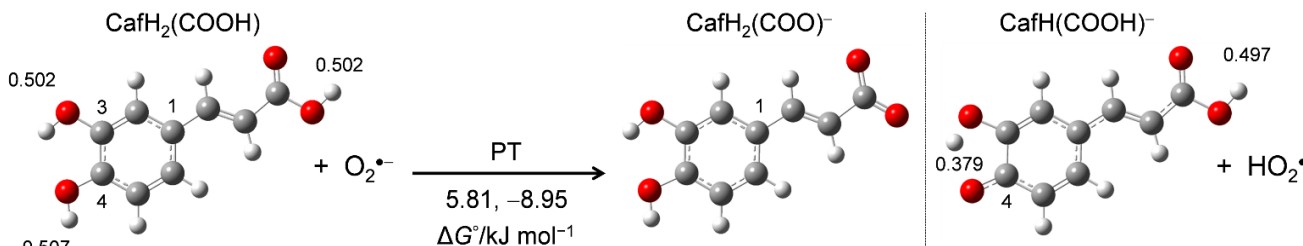

**Figure 4.** Optimized structures of CafH$_2$(COOH) and its deprotonated anion conformers (CafH$_2$(COO)$^-$ and CafH(COOH)$^-$) along the PT route to O$_2{}^{\bullet-}$ in DMF, calculated by DFT-B3LYP/PCM/6-311+G(3df,2p). The $\Delta G^\circ$s (kJ mol$^{-1}$, 298.15 K) of PT and the NBO-determined charges distributed on the three OH protons of CafH$_2$(COOH) are also indicated.

### 3.4. Change in HOMO-LUMO Energies during PCET between CafH$_2$(COOH) and O$_2{}^{\bullet-}$

Figure 5 shows changes in HOMO-LUMO (the singly occupied molecular orbital, SOMO) energies (Hartree/a.u.) during the PCET between CafH$_2$(COOH) and O$_2{}^{\bullet-}$ in DMF, supplemented by a frontier molecular-orbital analysis. After the initial PT, the reactant species CafH$_2$(COOH), O$_2{}^{\bullet-}$, CafH(COOH)$^-$, (CafH$_2$(COO)$^-$), and HO$_2{}^\bullet$ are involved in the solution. The SOMO energy of HO$_2{}^\bullet$ ($-0.3142$) is much lower than the HOMO energies of CafH$_2$(COOH) ($-0.2278$) and its anions (CafH(COOH)$^-$, $-0.1782$; CafH$_2$(COO)$^-$, $-0.2127$), indicating that the electron acceptor is HO$_2{}^\bullet$ rather than O$_2{}^{\bullet-}$. Figure 1a demonstrates that HO$_2{}^\bullet$ was scavenged, and thus, the electron donor is CafH(COOH)$^-$/CafH$_2$(COO)$^-$ where the downhill energy relationship during the ET is shown (bold red lines in Figure 5). These HOMO-LUMO changes occur during the PT between CafH$_2$(COOH) and O$_2{}^{\bullet-}$ forming CafH(COOH)$^-$/CafH$_2$(COO)$^-$ and HO$_2{}^\bullet$, implying that the initial reaction is PT. Next, the HOMO-LUMO relationship between the products is reversed after the subsequent ET (red dotted lines in Figure 5). However, since H$_2$O$_2$ formed after the second PT has a lower HOMO ($-0.2754$) than HO$_2{}^-$ ($-0.1648$), the reverse ET cannot proceed, so the ET direction is dominantly determined by the subsequent PT. Judging from the HOMO-LUMO relationship for O$_2{}^{\bullet-}$ scavenging by CafH$_2$(COOH), the net PCET involves two PTs and one ET.

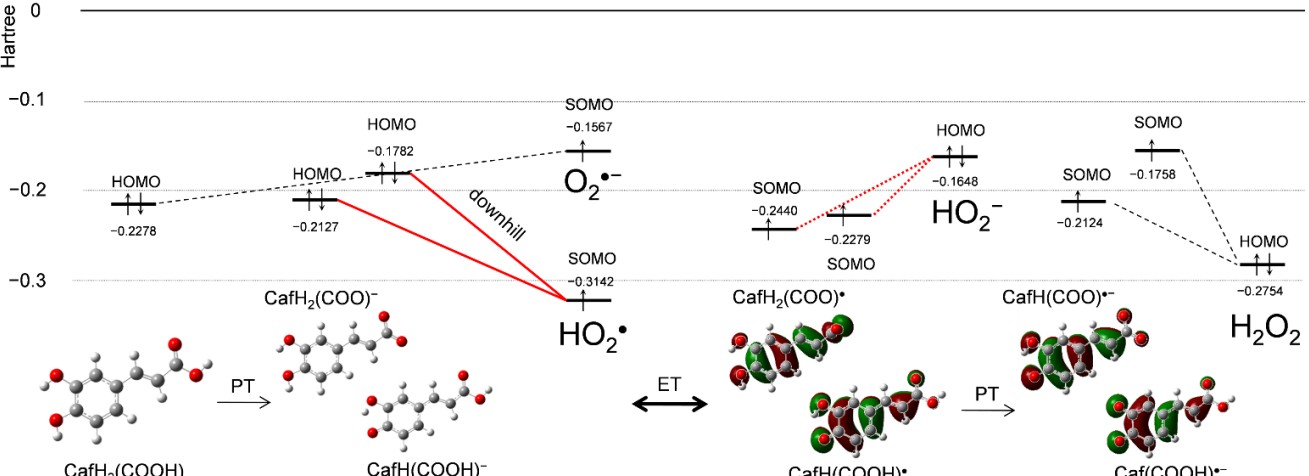

**Figure 5.** Changes in the HOMO−LUMO energies (Hartree/a.u.) along PCET mechanism between CafH$_2$(COOH) and O$_2{}^{\bullet-}$ and their corresponding chemical species in DMF, obtained by DFT at the (U)B3LYP/PCM/6-311+G(3df,2p) level.

### 3.5. Change in Free Energies during PCET between CafH$_2$(COOH) and O$_2^{\bullet-}$

Vibrational frequency calculations were performed to obtain changes in standard Gibbs free energy ($\Delta G°$/kJ mol$^{-1}$, 298.15 K) along the PCET in DMF. Figure 6 shows the equilibrium schemes and $\Delta G°$ of the six diabatic electronic states in the PCET involving two PTs and one ET between CafH$_2$(COOH) and O$_2^{\bullet-}$, and CafH$_2$(COO)$^-$ formed after the deprotonation of acryloyl OH and O$_2^{\bullet-}$. The main drivers of these sequential processes are the $\Delta G°$s of the individual reactions, the acid–base interaction and the redox potentials of the components. Since ET1 (a, 406.8; b, 346.7) is strongly endergonic, PT1 (a, −8.9; b, 16.2) dominantly occurs forming CafH(COOH)$^-$/CafH$_2$(COO)$^-$ and HO$_2^{\bullet}$. In the following pathway (the lower panels in Figure 6a,b), both PT3 (a, 302.9; b, 388.9) and ET2 (136.0, 6.5) are uphill endergonic, so the sequential PCET is unlikely to proceed. Efficient O$_2^{\bullet-}$ scavenging then requires a concerted PCET or HAT reaction involving one ET and one PT, corresponding to movement along the diagonal of the lower panels (red straight arrows). Notably, PT4 in Figure 6b is exergonic (−63.6), meaning that two PTs from the CatH$_2$ moiety must be coupled to one ET for successful O$_2^{\bullet-}$ scavenging. That is, PT from the acryloyl OH does not associate with ET. Then, another plausible pathway is a concerted ET and two PTs in one step after preforming the HB complexes between O$_2^{\bullet-}$ and the CatH$_2$ moiety of CafH$_2$(COOH)/CafH$_2$ (COO)$^-$ without generating high energy intermediates (red curved arrows). We refer to this pathway as the concerted two-proton-coupled electron transfer (2PCET) mechanism [16,29].

**Figure 6.** Six diabatic electronic states of PCET between (**a**) CafH$_2$(COOH) and O$_2^{\bullet-}$, and (**b**) CafH$_2$(COO)$^-$ and O$_2^{\bullet-}$ in DMF, involving two PTs and one ET. The $\Delta G°$s (kJ mol$^{-1}$, 298.15 K) of PT1–PT4 and ET1–ET3 were calculated using DFT at the (U)B3LYP/PCM/6-311+G(3df,2p) level.

For a comparative study, the $\Delta G°$s of the PCET of the other compounds were also calculated (Table 1). Thermodynamically, the total $\Delta G°$ of the net PCET obtained by summing the $\Delta G°$s of the two PTs and one ET is the energetic driving force. However, in case the PCET occurs along a pathway involving an infeasible single PT/ET, the total $\Delta G°$ cannot embody the energetic driving force. Along the plausible pathways, the concerted PCET (ET2–PT4/PT3–ET3) after the initial PT1, and 2PCET, was exergonic for all compounds. Therefore, the total $\Delta G°$ embodies the exergonic driving force similar to the case of CafH$_2$(COOH) (concerted, −41.3; total, −50.2). These $\Delta G°$s cannot explain the higher reactivities of CafH$_2$(COOH) and Et-CafH$_2$ than the other compounds toward electrogenerated O$_2^{\bullet-}$. In the electrochemical results (Figures 1 and 2), ferulic acid, isoferulic acid, and *p*-coumaric acid did not show O$_2^{\bullet-}$/HO$_2^{\bullet}$ scavenging ability, indicating that the CatH$_2$

moiety of CafH$_2$(COOH) and Et-CafH$_2$ confers the superior kinetics of the O$_2^{\bullet-}$ scavenging on CV time scales.

**Table 1.** $\Delta G^\circ$s (kJ mol$^{-1}$, 298.15 K) of PCET between O$_2^{\bullet-}$ and HCAs (CafH$_2$(COOH), ferulic acid, isoferulic acid, Et-CafH$_2$, and *p*-coumaric acid) in DMF, calculated using DFT at the (U)B3LYP/PCM/6-311+G(3df,2p) level.

| Compounds | PT1 | PT2 | PT3 | PT4 | ET1 | ET2 | ET3 | Concerted [1] | Total [2] |
|---|---|---|---|---|---|---|---|---|---|
| CafH$_2$(COOH) | −8.9 | −279.7 | 302.9 | −177.3 | 406.8 | 136.0 | −344.2 | −41.3 | −50.2 |
| Ferulic acid | 6.2 | −476.9 | 313.1 | −79.6 | 532.2 | 49.0 | −343.6 | −30.5 | −24.3 |
| Isoferulic acid | 4.5 | −471.2 | 298.2 | −77.1 | 517.0 | 41.2 | −334.1 | −35.9 | −31.4 |
| Et-CafH$_2$ | −5.4 | −365.8 | 371.3 | −85.9 | 402.9 | 42.5 | −414.6 | −43.3 | −48.7 |
| *p*-Coumaric acid | 7.7 | −353.7 | 301.9 | −79.4 | 420.1 | 58.6 | −322.8 | −20.8 | −13.1 |

[1] Concerted values are the summed $\Delta G^\circ$s of ET2 and PT4 (ET3 and PT3). [2] Total values are the summed $\Delta G^\circ$s of two PTs and one ET.

### 3.6. Potential-Energy Surfaces of the PCET between CafH$_2$(COOH) and O$_2^{\bullet-}$

To gain more insight into the PCET mechanism of O$_2^{\bullet-}$ scavenging by CafH$_2$(COOH) in DMF, we investigated the potential-energy surfaces with DFT and NBO analyses. The reaction is assumed to involve three elementary steps: (i) formation of the prereactive HB complex (PRC) from the free reactants (FRs), (ii) PCET reaction to the product complex (PC) via a TS, and (iii) dissociation of the PC yielding free products (FPs). First, we performed a geometry optimization of the stable HB complexes (PRC, intermediate complex, and PC) along the PCET reaction (Figure 7a). Then, optimized structures of the plausible PRC (CafH$_2$(COOH)–O$_2^{\bullet-}$) and PC (Caf(COOH)$^{\bullet-}$–H$_2$O$_2$) formed from the FRs and the FPs via two HBs (step i) were obtained. After optimization, the $\Delta G^\circ$ was reduced by 39.8 kJ mol$^{-1}$ (the $\Delta G^\circ$ of the PRC was set to zero in Figure 7a). Then, an energy profile ($\Delta G^\circ$, kJ mol$^{-1}$) along the IRC for the 2PCET, which forms the PC (step ii). The IRC shows that 2PCET occurs between the CatH$_2$ moiety of CafH$_2$(COOH) and O$_2^{\bullet-}$ in a one-kinetic process via a TS with low activation energy ($E_a$) at 53.8 kJ mol$^{-1}$ without generating any intermediates such as HO$_2^{\bullet}$, HO$_2^-$, CafH(COOH)$^-$, and CafH(COOH)$^{\bullet}$.

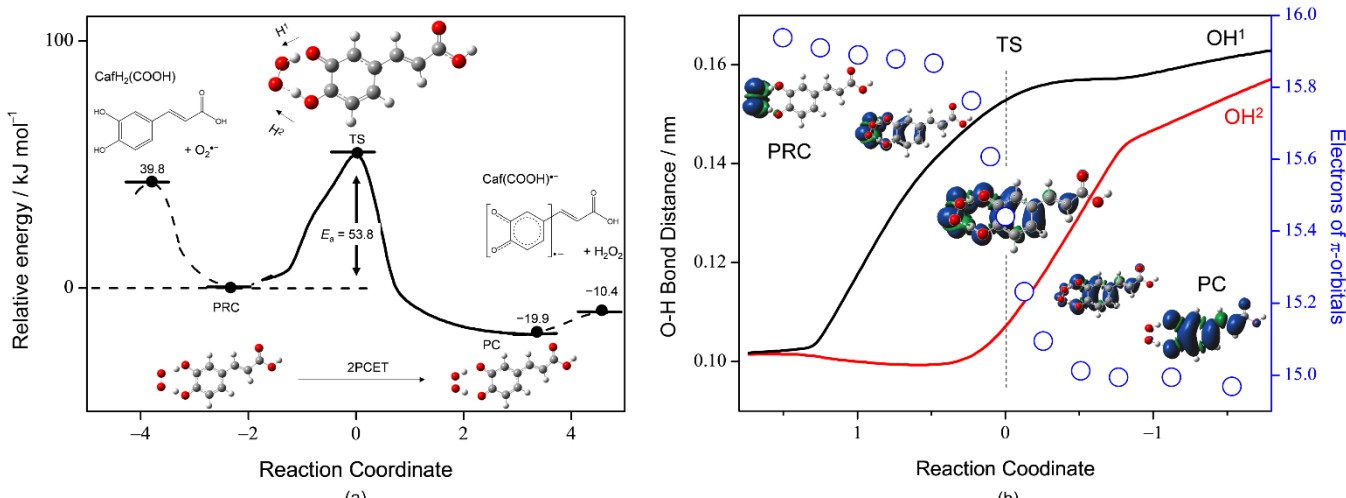

**Figure 7.** (**a**) Energy profile (kJ mol$^{-1}$) along the intrinsic reaction coordinate of the 2PCET between CafH$_2$(COOH) and O$_2^{\bullet-}$ in DMF, along with the structures of free reactants, prereactive HB complex (PRC), transition state (TS), product complex (PC), and free products (Caf(COOH)$^{\bullet-}$, H$_2$O$_2$). (**b**) Changes in OH bond distances (nm, left vertical axes; OH$^1$, black line; OH$^2$, red line) and the number of electrons (open circles) on the π-orbitals of CafH$_2$(COOH) (right vertical axes). All calculations were conducted by the DFT at the (U)B3LYP/PCM/6-311+G(3df,2p) level. Spin distributions and the electrons of π-orbitals were obtained by NBO analysis.

The dependence of OH bond distances (OH$^1$, black line; OH$^2$, red line) on the number of electrons in the $\pi$-orbital of the planar molecule of CafH$_2$(COOH) (blue circles) along the IRC are shown in Figure 7b. Spin density distributions localized on the radicals before and after the TS are also demonstrated, where the radical localized on O$_2^{\bullet-}$ in the initial PRC is transferred to Caf(COOH)$^{\bullet-}$ in the resulting PC. The changes in spin on the electron-donor side (CafH$_2$(COOH)) correlated well with the changes in the $\pi$-electrons of CafH$_2$(COOH). Furthermore, the changes in structures coupled with $\pi$-electron transfer occur simultaneously with sequential lengthening of the two OH bond distances (OH$^1$/OH$^2$) of the CatH$_2$ moiety. First, one phenolic proton (H$^1$) is attracted by O$_2^{\bullet-}$. At the TS, this attraction results in nearly complete deprotonation (OH$^1$) with one-half of the $\pi$-electrons transferred from CafH$_2$(COOH) to O$_2^{\bullet-}$. The second PT (H$^2$) accelerates the ET forward from the TS, eventually forming the PC.

For a comparative study, the potential-energy surfaces of the PCET with an IRC and TS between O$_2^{\bullet-}$ and the other substrates—CafH$_2$(COO)$^-$, Et-CafH$_2$, and CatH$_2$—were also investigated (Supplementary Materials, Figure S2). The 2PCET mechanisms were similarly mediated by the CatH$_2$ moiety and generated no intermediates. Table 2 lists the $\Delta G^\circ$ of FR, PC, and FP, and $E_a$ (the geometries of the TS are given in Tables S3–S5 of the Supplementary Materials). The $E_a$s of CafH$_2$(COOH) (53.8 kJ mol$^{-1}$), CafH$_2$(COOH)$^-$ (50.0 kJ mol$^{-1}$), Et-CafH$_2$ (53.2 kJ mol$^{-1}$), and CatH$_2$ (52.5 kJ mol$^{-1}$) were very similar and as low as the hydrogen-bonding energy. In Table 2, the relationship in which the difference in the $E_a$ values is proportional to the difference in their enthalpy of reaction shown as the $\Delta G^\circ$s of the PC suggests that the 2PCET mechanism of the same framework (between CatH$_2$ moiety and O$_2^{\bullet-}$) follows the Bell–Evans–Polanyi principle [31], justifying the conclusion that "CafH$_2$(COOH) scavenges O$_2^{\bullet-}$ through the 2PCET" and the methods adopted in this study. The $\Delta G^\circ$s of dissociating the PC to the FPs were $-10.4$ kJ mol$^{-1}$ for CafH$_2$(COOH), $-10.8$ kJ mol$^{-1}$ for CafH$_2$(COO)$^-$, and $-10.0$ kJ mol$^{-1}$ for Et-CafH$_2$, 55–56 kJ mol$^{-1}$ lower than that for CatH$_2$ (45.3 kJ mol$^{-1}$). Thus, the superior O$_2^{\bullet-}$-scavenging ability of CafH$_2$(COOH) is mainly attributable to the $\Delta G^\circ$s of dissociating the PC (step iii) that promote the net PCET reaction, rather than to the kinetics of 2PCET via the TS (step ii).

**Table 2.** $\Delta G^\circ$ and $E_a$ values (kJ mol$^{-1}$, 298.15 K) of the 2PCET between O$_2^{\bullet-}$ and the substrates (CafH$_2$(COOH), CafH$_2$(COO)$^-$, Et-CafH$_2$, and CatH$_2$) in DMF, calculated using DFT at the (U)B3LYP/PCM/6-311+G(3df,2p) level.

| Reactants [1] | FR | TS ($E_a$) | PC | FP |
|---|---|---|---|---|
| CafH$_2$(COOH) (+O$_2^{\bullet-}$) | 39.8 | 53.8 | $-19.9$ | $-10.4$ |
| CafH$_2$(COO)$^-$ (+O$_2^{\bullet-}$) | 30.1 | 50.0 | $-27.6$ | $-10.8$ |
| Et-CafH$_2$ (+O$_2^{\bullet-}$) | 38.7 | 53.2 | $-20.3$ | $-10.0$ |
| CatH$_2$ (+O$_2^{\bullet-}$) | 71.6 | 52.5 | $-20.9$ | 45.3 |

[1] $\Delta G^\circ$s (kJ mol$^{-1}$) of PRC were set as a zero point.

Collectively, DFT results clarified that scavenging of O$_2^{\bullet-}$ by CafH$_2$(COOH) in DMF is governed by 2PCET involving two PTs and one ET in the formed PRC via two HBs, which corresponds to moving along the red curved arrows in Figure 6a. Figure 8 shows the net mechanism of O$_2^{\bullet-}$ scavenging by CafH$_2$(COOH) in DMF. In the 2PCET mechanism, ET occurs between the $\pi$ orbitals of oxygen orthogonal to the molecular frameworks of the donor and acceptor, and PT occurs between the $\sigma$ orbitals of oxygen along the HBs [16]. The numbers of spin distributed on the oxygen species (O$_2^{\bullet-}$, H$_2$O$_2$), the CatH$_2$ moiety, and the acryloyl group, along step iii, the dissociation of the PC (9.5 kJ mol$^{-1}$), are also given in Figure 8. Notably, the spins are distributed on the acryloyl group of the PC (0.079) and on the dissociated Caf(COOH)$^{\bullet-}$ (0.110) that expands the $\pi$-conjugated plane, demonstrating that the radical product (PC and Caf(COOH)$^{\bullet-}$) is more stabilized by the acryloyl group than the orthoquinone radical generated from CatH$_2$. These results are in good correlation with the $\Delta G^\circ$s of dissociating PC to FP (Table 2), suggesting that the acryloyl group of

CafH$_2$(COOH), CafH$_2$(COO)$^-$, and Et-CafH$_2$, thermodynamically promotes step iii, the dissociation of PC. These results imply that the acryloyl group play a promoting role in the net 2PCET mechanism (Figure 8) between O$_2^{\bullet-}$ and the catechol moiety via the dissociation (step iii) for efficient O$_2^{\bullet-}$ scavenging abilities of CafH$_2$(COOH).

**Figure 8.** Net PCET mechanism between CafH$_2$(COOH) and O$_2^{\bullet-}$ in DMF with the $\Delta G^\circ$ and $E_a$ values (kJ mol$^{-1}$, 298.15 K). Step (i) initial formation of PRC; step (ii) concerted 2PCET; step (iii) dissociation of the PC. $\Delta G^\circ$ and $E_a$ were calculated using the DFT-(U)B3LYP/PCM/6-311+G(3df,2p) method. The numbers are spins distributed on the oxygen species (O$_2^{\bullet-}$, H$_2$O$_2$), the CatH$_2$ moiety, and the acryloyl group obtained by NBO analysis.

## 4. Conclusions

In the present study, we investigated the reactivities of HCAs involving CafH$_2$(COOH) toward the electrogenerated O$_2^{\bullet-}$ in DMF. In the CV and spectral measurements, it was confirmed that O$_2^{\bullet-}$ was successfully scavenged by CafH$_2$(COOH) and Et-CafH$_2$ through PCET mediated by the CatH$_2$ moiety over the time scale of the CV measurements. Then, DFT calculations clarified that the concerted 2PCET mechanism involving two PTs and one ET via the CatH$_2$ moiety (not involving PT from the acryloyl group) embodies the superior kinetics of O$_2^{\bullet-}$ scavenging by CafH$_2$(COOH) and Et-CafH$_2$. The net mechanism involves the initial formation of PRC followed by 2PCET and dissociation of the PC into FPs. The acryloyl group contributes to the thermodynamic stability of the product radical, promoting the dissociation of the PC. Efficient O$_2^{\bullet-}$ scavenging abilities of CafH$_2$(COOH) and Et-CafH$_2$ are derived from both the CatH$_2$ moiety and the acryloyl group.

Although the presented results are specific to chemical reactions in aprotic DMF solvent, the PCET theory is adaptable to biological processes in biotic structures such as lipid bilayers. Therefore, we hope that the findings will reveal the mechanistic actions of O$_2^{\bullet-}$ scavenging by HCAs and the health benefits of CafH$_2$(COOH), thus securing its pharmacological use as a phytoalexin.

**Supplementary Materials:** The following is available online at https://www.mdpi.com/article/10.3390/electrochem3030024/s1, Table S1: CV parameters, Figure S1: In situ electrolytic ESR/UV-vis system, Table S2: Optimized geometry of caffeic acid, Figure S2: Energy profiles along IRC of 2PCET between CafH$_2$(COO$^-$) and O$_2^{\bullet-}$, Tables S3–S6: Optimized geometry of TS.

**Author Contributions:** Conceptualization, T.N.; methodology, B.U.; resources, T.N.; writing—original draft preparation, T.N.; writing—review and editing, B.U. All authors have read and agreed to the published version of the manuscript.

**Funding:** This research was funded by Grant-in-Aid for Scientific Research, grant number 19K16338 from Japan Society for the Promotion of Science (JSPS).

**Institutional Review Board Statement:** Not applicable.

**Informed Consent Statement:** Not applicable.

**Data Availability Statement:** Data available in a publicly accessible repository.

**Conflicts of Interest:** The authors declare no conflict of interest.

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
