# Peer review of "Reactivities of Hydroxycinnamic Acid Derivatives Involving Caffeic Acid toward Electrogenerated Superoxide in N,N-Dimethylformamide"

_2673-3293, doi:10.3390/electrochem3030024_

Round 1
Reviewer 1 Report
This is a nicely written article brining new experimental and theoretical information concerning the measurement of reactive oxidative species. I suggest acceptance of the paper for publication in a leading international journal.
Author Response
Thank you for the acceptance.
Reviewer 2 Report
Comment:
This manuscript reports a comprehensive study on the electro-redox reaction of five different hydroxycinnamic acid derivatives. The authors apply the in situ electrochemical ESR and UV-vis experiment and identify the radical intermediates during the electro-redox reactions. Also, the authors use DFT calculation to further explain the charge transfer mechanism and reactivity of the hydroxycinnamic acid derivatives. However, there are some remaining questions in this manuscript including the electrode choice and the peak shift. Thus, I think a minor revision is needed for this manuscript before it published on Electrochem at this stage.
Specific comments:
1. The authors performed the electrochemical measurement with a glassy carbon RDE as working electrode for CVs and the in situ ESR and UV-vis measurement with Pt wire and Pt mesh. Will the difference in working electrodes affect the reaction?
2. In Figure 1, the reduction peaks of five organic acids shift differently with the concentration increase. (c) has a positive shift with the increase of concentration while (e) has a negative shift. The other three (a) (b) and (d) seem to have no shift. Is there any explanation for the peak shift of these CV scans?
3. To follow up the previous question, the signal gains of these five organic acids are also different. Is it caused by experiment setup or the different intrinsic electrochemical activities of different chemical structure?
4. The authors provide a computational result on the charge transfer between CafH2(COOH) and O2•−. What is the concentration of O2•− used for the DFT calculation? Is the concentration of O2•− quantifiable in experiment?
Technical Comments:
5. In the figure 1, the CVs of different concentration are overlapped in thin dash line and hard for readers to understand the trend. A optimized line thickness or line style is needed.
6. The authors used the ‘homemade’ in situ electrolytic ESR and UV-vis spectral system. However, more detailed parameters of these two systems are necessary to improve the reproducibility of this work. For example, what is the electrode area of the OTTLE? What is the length of the Pt wire in electrolytic ESR system?
Reviewer 3 Report
"Reactivities of Hydroxycinnamic Acid Derivatives Involving Caffeic Acid toward Electrogenerated Superoxide in N,N-Dimethylformamide through Proton-coupled Electron Transfer"
The Authors present and discuss results on the scavenging of the superoxide radical anion (O2•−) by (2E)-3-(3,4-dihydroxyphenyl)prop-2-enoic acid, investigated by cyclic voltammetry. The manuscript is well written, however some issues need to addressed prior to publication.
* Title: meaningful.
* Keywords: meaningful.
* Abstract: meaningful, yet the Authors need to further emphasize on the novelty and importance of their work; use present tense (recommended) and avoid redundant text.
- Introduction
* this section is too short, missing important information; the Authors need to insist more on the novelty and importance of their approach with respect to literature and their previous work (some important publications are missing): further explain on your approach, and also consider providing more / further references to it.
* the last paragraph of the section needs to be a brief presentation of the work herein, with sufficient and relevant details; please rephrase this section accordingly.
- Materials and Methods
* as a general remark, this section and subsections are too short (incomplete) and the experimental methods need to be further discussed; however, there's no need to present well-known techniques (references will suffice, please provide more where available); avoid redundant text; provide further information on the compounds, and their full names prior to using the acronyms.
- Results and discussion
* please start the section by describing the main aspects of your work - what do you seek and what is your plan in doing so (a few phrases will suffice).
* as a general overview / remark to this section: the Authors need to further discuss their results in a more correlated manner; provide more references to sustain your results (where available).
* please further present and discuss in text all figures and tables, provide references to sustain your results (where available); figures and tables need to be in text, not after.
* a final (last) paragraph of section 3 must be included, to provide the reader with a brief conclusion of your work / manuscript (and insist more on the novelty of your approach).
- Conclusion
* this section is poorly written and the Authors need to emphasize more on the novelty and importance of this approach, providing sufficient data relevant to this study; avoid bullet points (recommended).
To conclude, the manuscript should be considered for publication only after careful revision.
Reviewer 4 Report
In this study, the authors explored the mechanistic insights into the structural features of CafH2(COOH), and the reaction between electrogenerated O2•- and HCA derivatives involving CafH2(COOH) utilizing density functional theory (DFT). Additionally, the authors also conducted an electrochemical investigation to study the scavenging mechanism of O2•- and PT-forming HO2• for CafH2(COOH).
I recommend publishing this after the following points are considered.
1. The activation energy and enthalpy have a linear connection, according to classical thermodynamics and the Bell-Evans-Polanyi principle. However, many systems do not follow this concept. You should justify your method and conclusion that “CafH2(COOH) scavenges O2•− through a 2PCET involving concerted two PTs and one ET” by pointing out the relationship between reaction and activation energy, which is known as the Bell-Evans-Polanyi principle.
2. While PCET reactions generate the most energy-efficient pathway, they may raise the kinetic barrier to the entire process. So, while this study discussed the thermodynamic feasibility of the PCET between CafH2(COOH) and O2•, it made no mention of the kinetic feasibility of the process.
Round 2
Reviewer 3 Report
"Reactivities of Hydroxycinnamic Acid Derivatives Involving Caffeic Acid toward Electrogenerated Superoxide in N,N-Dimethylformamide through Proton-coupled Electron Transfer" - revision 1
The Authors have correctly addressed most of the issues raised during the peer review procedure. The manuscript is now suitable for publication.